# Prior antibiotics and risk of subsequent Herpes zoster: A population-based case control study

**David Armstrong**[1]*, **Alex Dregan**[2]*, **Mark Ashworth**[1], **Patrick White**[1]

1 School of Life Course and Population Sciences, King's College London, London, United Kingdom,
2 Department of Psychological Medicine, Institute of Psychiatry, Psychological and Neurosciences, King's College London, London, United Kingdom

* David.armstrong@kcl.ac.uk (DA); alexandru.dregan@kcl.ac.uk (AD)

**Data Availability Statement:** The data underlying the results presented in the study are available from the UK's Clinical Practice Research Datalink. Permission to access the data must be sought and

## Abstract

### Background

The effect of antibiotics on the human microbiome is now well established, but their indirect effect on the related immune response is less clear. The possible association of Herpes zoster, which involves a reactivation of a previous varicella zoster virus infection, with prior antibiotic exposure might indicate a potential link with the immune response.

### Methods

A case-control study was carried out using a clinical database, the UK's Clinical Practice Research Datalink. A total of 163,754 patients with varicella zoster virus infection and 331,559 age/sex matched controls were identified and their antibiotic exposure over the previous 10 years, and longer when data permitted, was identified. Conditional logistic regression was used to identify the association between antibiotic exposure and subsequent infection in terms of volume and timing.

### Results

The study found an association of antibiotic prescription and subsequent risk of varicella zoster virus infection (adjusted odds ratio of 1.50; 95%CIs: 1.42–1.58). The strongest association was with a first antibiotic over 10 years ago (aOR: 1.92; 95%CIs: 1.88–1.96) which was particularly pronounced in the younger age group of 18 to 50 (aOR 2.77; 95%CIs: 1.95–3.92).

### Conclusions

By finding an association between prior antibiotics and Herpes zoster this study has shown that antibiotics may be involved in the reactivation of the varicella zoster virus. That effect, moreover, may be relatively long term. This indirect effect of antibiotics on viruses, possibly mediated through their effect on the microbiome and immune system, merits further study.

a fee is likely to be payable. These data are collected by CPRD from primary care electronic health records and as such are third party data neither collected nor owned by the authors. Application to access the database can be made here: https://www.cprd.com/Data-access. An application would involve submitting a protocol that would have to be approved by the Research Governance Process. This study was approved and accessed under protocol number 20_134. An access fee is also charged. The authors had no special access privileges though as their institution subscribes to a licence no fee was payable for this particular study. As these are data containing sensitive patient data, the deidentification of records is important. The Research Governance Process ensures that protocols are ethically sound and approval is given only for requested data.

**Funding:** The author(s) received no specific funding for this work.

**Competing interests:** The authors have declared that no competing interests exist.

## Introduction

Herpes zoster virus infection (HZ) is caused by reactivation of the varicella zoster virus that lies dormant in the dorsal root ganglia after an episode of chickenpox, usually experienced in childhood. A recent meta-analysis of 88 studies involving nearly 4 million cases of HZ [1, 2] confirmed earlier observations that the most important risk factor was immunosuppression, sometimes linked to cancer treatments. Age, family history and physical trauma are also associated with HZ [3] and, to a lesser extent, some co-morbidities such as diabetes, rheumatoid arthritis, cardiovascular diseases, renal disease, systemic lupus erythematosus, and inflammatory bowel disease [4–6]. Yet given the frequency of HZ in the general population a common conclusion of HZ studies is that its key attributable causes remain unknown [1, 7].

Existing risk factors and laboratory studies have shown a clear relationship between HZ and cell-mediated immunity but what triggers that immune reaction for most patients remains unclear. One possibility is a disturbance in the patient's microbiome [8]. The microbiome is known to be intimately linked to the development and maintenance of immune responses including systemic effects such as atopy [9, 10]. Microbiome dysbiosis may therefore play a part in susceptibility to varicella zoster virus reactivation, a process that may occur when a patient is given broad spectrum antibiotics for some unrelated condition. It is now well established that antibiotics create some degree of dysbiosis in the microbiome [11–13] and there is also some evidence that, in its turn, alterations of the gut microbiome may cause dysregulated mucosal immune responses leading, for example, to the onset of inflammatory bowel diseases [14]. To explore the effect of antibiotics on the reactivation of varicella zoster virus we therefore carried out a case control study comparing the antibiotic prescriptions of patients with an HZ diagnosis and those without.

## Method

A case-control study was implemented in one of the world's largest primary care database, the Clinical Practice Research Datalink (CPRD). The CPRD currently contains medical records from 1,498 general practices (19% of UK practices) capturing detailed clinical, therapeutic, lab test, immunisation, referral, and lifestyle data for over 39 million patients (~13 million active patients). CPRD has been granted generic ethics approval by the UK's Health Research Authority for observational studies that make use of only anonymised data and linked anonymised NHS health-care data (ref. 21/EM/0265). The ethics committee waived the need for individual patient consent for use of their deidentified data. The mean follow-up of currently active patients is about 12 years. Data are collected from contributing practices on a daily basis and processed to create monthly snapshots for clinical and epidemiological investigations [15]. The organisation of the healthcare system in the UK (a publicly funded health service) where general practitioners are considered the 'gatekeepers' and coordinators of health care makes these data a valuable resource for epidemiological investigations. CPRD primary care practices are broadly representative of the UK population [16]. The data have been extensively validated for pharmaco-epidemiological, clinical, and health service usage research [17–19]. The Aurum version of the CPRD database was used for this study [15].

### Study population

Cases were patients aged ≥18 years (HZ is rare among children) with a recorded diagnosis of HZ between 1ˢᵗ January 2008 and 31ˢᵗ December 2018 using SNOMED/Read diagnostic codes (see S1 Appendix). The index date was defined as the first date that a diagnosis of HZ was ever recorded in a patient medical record. Cases and controls had to have a minimum of 12-months of medical history in CPRD prior to the index date and to have no history of HZ at cohort

entry. Two controls were sought for each case, individually matched on age (within 5 years to ensure sufficient matched controls per practice), gender, and family practice; controls were given the index date of the HZ diagnosis of their matched case. Incidence density sampling was used to select controls to minimise differential loss to follow-up. We excluded patients with a HZ vaccination prior to HZ index date (~7% of cases and 6% of controls) to limit the potential for selective bias as the vaccination was introduced in 2013 in the UK and is restricted to patients aged 70 to 78 years. Data were extracted in February 2021.

## Exposure

The exposure variable included any antimicrobial prescription (excluding anti-tuberculous and anti-leprotic drugs) prior to the HZ index date. The class of a given antibacterial prescription was derived from subchapters of the British National Formulary (chapter 5.1) and included penicillins, cephalosporins, tetracyclines, sulphonamides, trimethoprim, metronidazole and quinolones. Information extracted included the class of antimicrobial prescriptions, the number of antimicrobial prescriptions issued during different follow-up periods (0, 1, 2, 3, 4, 5, 6–10, >10 years) and the interval between the first antimicrobial prescription and the HZ index date as measures of the extent of exposure.

## Confounders

Several variables that have been associated with HZ risk and antibiotic prescribing were included as covariates [2, 5]. These included matching variables (age, gender, and practice); body mass index (BMI), ($<18.5$, 18.5 to 25, $>25$ to $<30$, 30 to $<35$, and $\geq 35$ kg/m$^2$); lifestyle factors such as smoking (never, ex-smoker, current smoker) and alcohol (never, ex-drinker, current drinker); comorbidity including depression, HIV infection, cancer, renal disease, autoimmune disorders (including rheumatoid arthritis, systemic lupus erythematosis, inflammatory bowel disorders), chronic obstructive pulmonary disease (COPD), type 2 diabetes, cardiovascular diseases (including stroke and heart failure); and co-prescribing within three months from index date for HZ including use of immunosuppressive therapies, corticosteroids, non-steroidal anti-inflammatory drugs, and disease-modifying anti-rheumatic drugs (DMARDs). For comorbidities, body mass index (BMI) and lifestyle factors, the value closest to the index date for HZ was included.

## Statistical analysis

Our analyses were restricted to patient-level data from the CPRD Aurum database. Descriptive analyses (e.g. frequencies, means) for baseline covariates and explanatory variables were conducted for cases and controls. Data were collected from the earlier of date of registration with the general practice or the date of first antibiotic prescription and continued until the index date for HZ (case index date for matched controls). Conditional logistic regression was used to determine the odds ratios (ORs) and 95% confidence intervals (95%CIs) for incident HZ associated with previous antimicrobial exposure. Due to the matching of case with controls, all analyses accounted for age, gender, calendar time and family practice. The analyses further adjusted for the main hypothesised confounding variables described above. Prescriptions issued 12 months before the diagnosis were excluded from the analysis to reduce the risk of protopathic bias given that early presentations of HZ might have been misdiagnosed and antibiotics prescribed; a conservative 12 month exclusion period was chosen. Sensitivity analyses were carried out that included this period.

Separate estimation models were conducted for: number of prescriptions compared to no prescriptions; the interval in years since the first ever prescription before the index date

compared to no prescribing; the age group of the patients and the number of prescriptions; and antibiotic class (e.g. penicillins, sulfonamides, etc) with the risk of HZ. We used the likelihood ratio test to assess potential temporal variation in the odds of HZ after first antibiotic prescription. These tests were repeated for each class of antibiotics. Interaction between age at HZ diagnosis and antibiotic prescribing was assessed by including interaction parameters between age at index date (<50, 50–64, 65–69, 80+) and antibiotic prescription (ever) using the likelihood ratio test to assess the strength of the evidence of age interaction (e.g. comparing models with and without the interaction parameters). Participants with missing data on any explanatory variables or covariates were excluded from the analyses (complete case analysis), which is valid if missingness is independent of the outcome (HZ), conditional on all covariates [20].

We have performed additional sensitivity analysis to validate the findings from the complete-case analyses by using multiple imputation with chained equation (10 imputed datasets). Under the missing at random assumption, multiple imputation is superior to complete-case analysis [21]. As these analyses validated complete-case analysis, we are presenting only the latter findings here. Following Rothman [22] and Ridker et al. [23] the analyses did not adjust for multiple comparisons. We used the metan command (with random effects) in Stata to estimate the overall effect size and related heterogeneity statistic ($I^2$) for HV across the different classifications of the study exposure (antibiotics) within each estimation model. Data were analysed using STATA version 15 (StataCorp, College Station, TX).

## Results

A total of 163,754 patients with HZ and 331,559 age/sex matched controls were identified in CPRD (Fig 1). Table 1 shows the age and sex matching together with the distribution of covariates that were used for adjusting the logistic regression models.

Fig 2 shows the risk of HZ is partly related to the number of antibiotics prescribed over the previous 10 years though beyond the first one or two prescriptions risk did not increase markedly as indicated by the overlapping confidence intervals (for two prescriptions the aOR was 1.45: 95%CIs 1.40–1.49; for over ten prescriptions the aOR was 1.62: 95%CIs 1.47–1.72). Sensitivity analyses that included prescriptions issued during the 1 year protopathic exclusion period showed similar results. The association of antibiotic prescription with HZ increased the earlier a first prescription was recorded (Fig 3). The highest aOR of 1.92 (95%CIs 1.88–1.96) was for a first prescription issued over ten years earlier.

Fig 4 shows the relationship between number of prescriptions and HZ broken down by age groups. The largest associations were observed in the youngest group, those aged 18–50 years (aOR 2.13: 95%CIs 1.86–2.39), the smallest in the oldest group, over 65 years (aOR 1.34: 95% CIs 1.28–1.40). Fig 5 shows the association of HZ with different antibiotic types. Since the findings suggested that models without the interaction parameters between age at index date and antibiotic prescription provided a better fit to the data, we present these results here. The association between antibiotic type and HZ showed some variability from tetracyclines with an aOR of 1.30 (95%CIs 1.22–1.37) to penicillins with an aOR of 1.87 (95%CIs 1.84–1.90). According to the $I^2$ statistic, there was significant heterogeneity (p<0.001) between the type of antibiotics and the HZ.

## Discussion

This study has shown an association of antibiotic prescription and subsequent risk of HZ. That association was primarily related to one or two prescriptions with a first prescription more than 10 years prior to the HZ episode carrying the highest risk. Younger patients seemed more at risk than older though incidence is higher in the latter group.

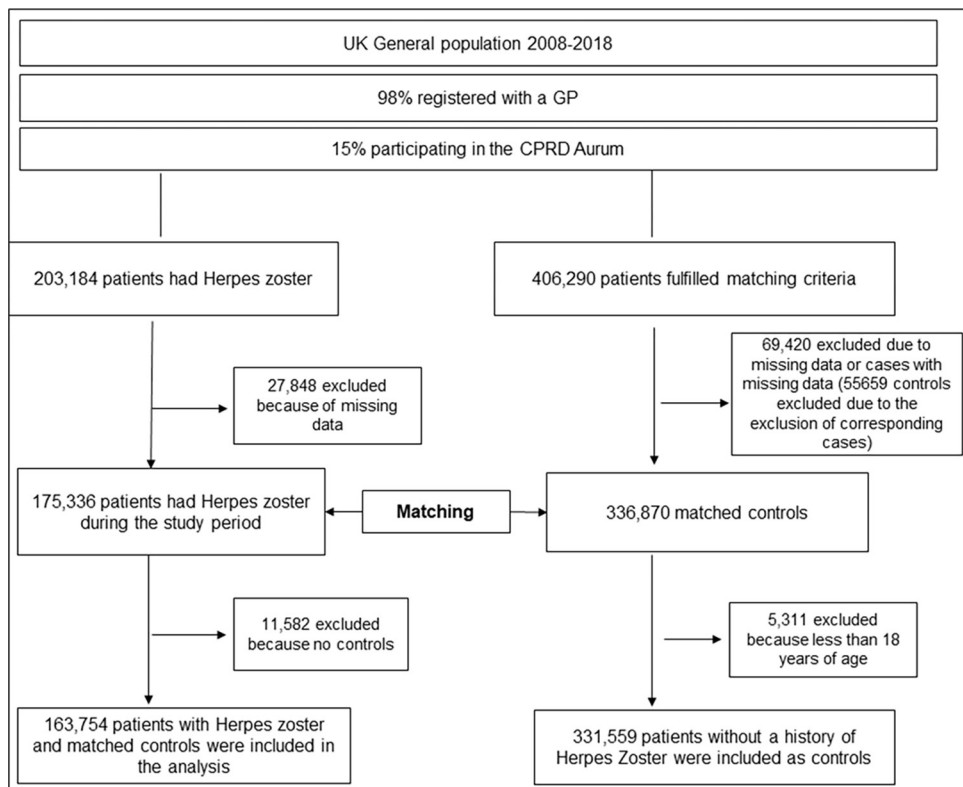

**Fig 1. Flow diagram of study recruitment.**

Several studies, albeit with small numbers of patients and limited follow-up, have examined the long-term impact of antibiotics on the microbiome in adults. They generally find some recovery of the microbiome within a year of exposure but to a new and different stable state [24–26]. The consequences of this change in adults are largely unknown [27]. There is, however, some evidence that exposure to antibiotics in infancy significantly changes the composition of the microbiome leading to long-term consequences for host susceptibility to viral and bacterial infections [28] and other diseases [29]. Further, a study of the relationship between antibiotics and rheumatoid arthritis, an autoimmune disease, also showed an increased risk several years after taking antibiotics [30]. Results from the present study might indicate longer term consequences for adults of antibiotic perturbation than has been recognised to date.

This study could not examine biological mechanisms that might underpin the association between antibiotics and varicella zoster virus reactivation but there is growing evidence that the increased risk might be caused by microbiome dysbiosis and its subsequent effect on the immune response [31–33]. The fact that antibiotics have a significant impact on the bacterial microbiome is well established [24, 34–36]; their effect on viruses, however, is less clear. The gut microbiome has been shown to associated with immune cell dynamics in humans [37] and immune-mediated side-effects are recognised as a short-term reaction to antibiotics [38] as well as other drugs [39]. Animal studies have suggested that the microbiome regulates virus-specific CD4 and CD8 T cells and antibody responses following respiratory influenza virus infection [40].

There is considerable evidence that the immune response declines with age [41, 42] yet in this study the patients at greater risk were those in younger groups. While HZ is more

**Table 1. Distribution of covariates by Herpes zoster cases and controls.**

| | Herpes zoster | Controls |
|---|---|---|
| | N = 163,754(%) | N = 331,559(%) |
| **Matching variables** | | |
| Gender | | |
| Female | 94 561(58) | 191 964(58) |
| Male | 69 193(42) | 139 595(42) |
| Age–Mean ± SD | 62(17) | 61(17) |
| **Adjusted for variables** | | |
| Cancer | 12 724(8) | 22 259(7) |
| Depression | 39 344(24) | 75 340(23) |
| Systemic lupus erythemathosus | 509(<1) | 691(<1) |
| Rheumatoid arthritis | 4 908(3) | 6 714(2) |
| Inflammatory bowel disorders | 9 993(6) | 14 476(4) |
| Cardiovascular diseases | 19 501(12) | 34 926(11) |
| Diabetes mellitus | 17 288(11) | 33 913(10) |
| HIV infection | 148(<1) | 240(<1) |
| Chronic kidney disease | 19 933(12) | 33 603(10) |
| Liver disease | 2 486(1) | 4 608(1) |
| Smoking status | | |
| Current smoker | 47 744(29) | 88 319(27) |
| Ex-smoker | 21 412(13) | 52 182(16) |
| Never smoker | 92 838(57) | 175 336(53) |
| Missing | 1760(1) | 15 722(5) |
| Drinking status | | |
| Current drinker | 9 025(6) | 21 356(6) |
| Ex-drinker | 2 094(1) | 4 009(1) |
| Never drinker | 152 635(93) | 306 194(93) |
| Missing | | |
| BMI category | | |
| Normal weight | 49 560(30) | 97 896(30) |
| Underweight | 2 715(2) | 6 474(2) |
| Overweight | 54 149(33) | 100 564(30) |
| Obese | 25 858(16) | 47 319(14) |
| Morbidly obese | 13 393(8) | 24 835(7) |
| Missing | 18 079(11) | 56 471(16) |
| Immunosuppressive therapy | 486(<1) | 564(<1) |
| Corticosteroid therapy | 19 758(12) | 28 893(9) |
| NSAID | 51 854(32) | 90 154(27) |

Abbreviation: BMI-body mass index; HIV–human immunodeficiency virus; NSAID- non steroidal anti-inflammatory drugs.

common in older patients, this study suggests that younger patients may be more susceptible to antibiotic damage despite stronger immune systems. If that strength derives from immuno-plasticity this might increase vulnerability to microbiome damage by antibiotics.

Finally, the study showed some variability with the type of antibiotic prescribed. Different antibiotics are known to affect the bacterial microbiome in different ways [36, 43] but less is known about whether this extends to the immune response. Tetracyclines have been shown to offer some degree of immune protection resulting in mucosal healing [44]; in this study they

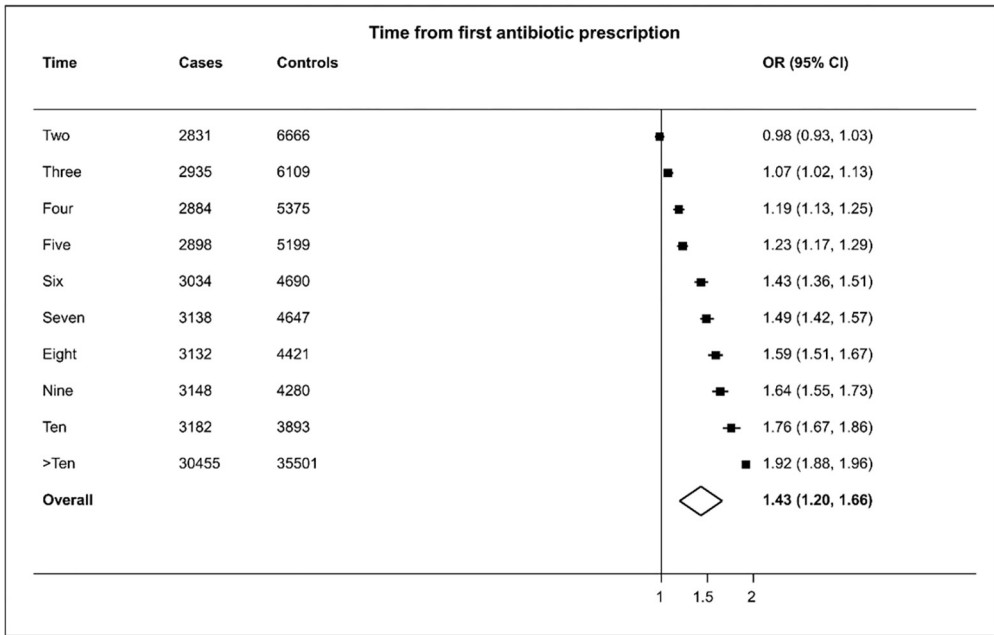

| Total antibiotics over the previous 10+ years | | | | |
|---|---|---|---|---|
| Number | Cases | Controls | | OR (95% CI) |
| One | 21768 | 33933 | | 1.34 (1.31, 1.37) |
| Two | 8621 | 12236 | | 1.45 (1.40, 1.49) |
| Three | 4202 | 5648 | | 1.51 (1.45, 1.58) |
| Four | 2369 | 3177 | | 1.51 (1.42, 1.60) |
| Five | 1512 | 1947 | | 1.57 (1.46, 1.70) |
| Six/Ten | 2631 | 3354 | | 1.57 (1.49, 1.68) |
| >Ten | 1253 | 1544 | | 1.62 (1.47, 1.72) |
| Overall | | | | 1.50 (1.42, 1.58) |

**Fig 2. Results of adjusted logistic regression showing association of HZ with the number of antibiotic prescriptions over the previous 10+ years.**

had one of the smaller associations with HZ. The commoner group of penicillins had the largest associations.

The main strength of the study is the large number of patients identified with HZ and their controls together with the quality of clinical data in CPRD. Most cases of HZ will be taken to the primary care practitioner in the UK and the diagnosis recorded. Prescription recording is also comprehensive as close to 99% are issued electronically. CPRD, however, does not record

| Time from first antibiotic prescription | | | | |
|---|---|---|---|---|
| Time | Cases | Controls | | OR (95% CI) |
| Two | 2831 | 6666 | | 0.98 (0.93, 1.03) |
| Three | 2935 | 6109 | | 1.07 (1.02, 1.13) |
| Four | 2884 | 5375 | | 1.19 (1.13, 1.25) |
| Five | 2898 | 5199 | | 1.23 (1.17, 1.29) |
| Six | 3034 | 4690 | | 1.43 (1.36, 1.51) |
| Seven | 3138 | 4647 | | 1.49 (1.42, 1.57) |
| Eight | 3132 | 4421 | | 1.59 (1.51, 1.67) |
| Nine | 3148 | 4280 | | 1.64 (1.55, 1.73) |
| Ten | 3182 | 3893 | | 1.76 (1.67, 1.86) |
| >Ten | 30455 | 35501 | | 1.92 (1.88, 1.96) |
| Overall | | | | 1.43 (1.20, 1.66) |

**Fig 3. Results of adjusted logistic regression showing association of HZ with the number of years since first antibiotic prescription.**

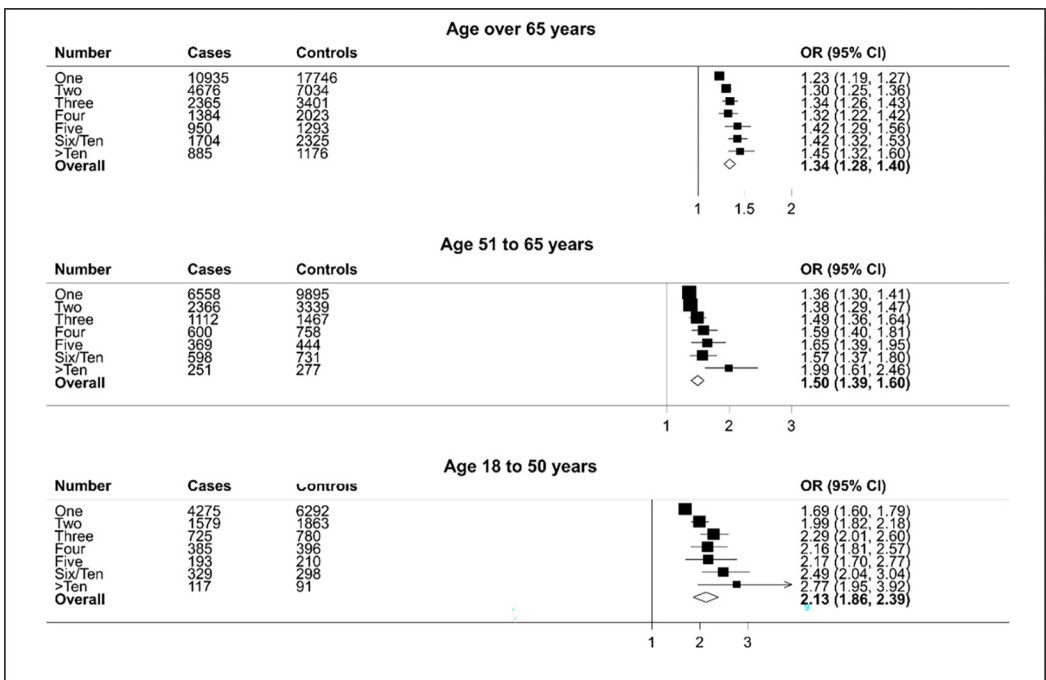

**Fig 4. Results of adjusted logistic regression showing association of HZ with the number of antibiotic prescriptions by age group.**

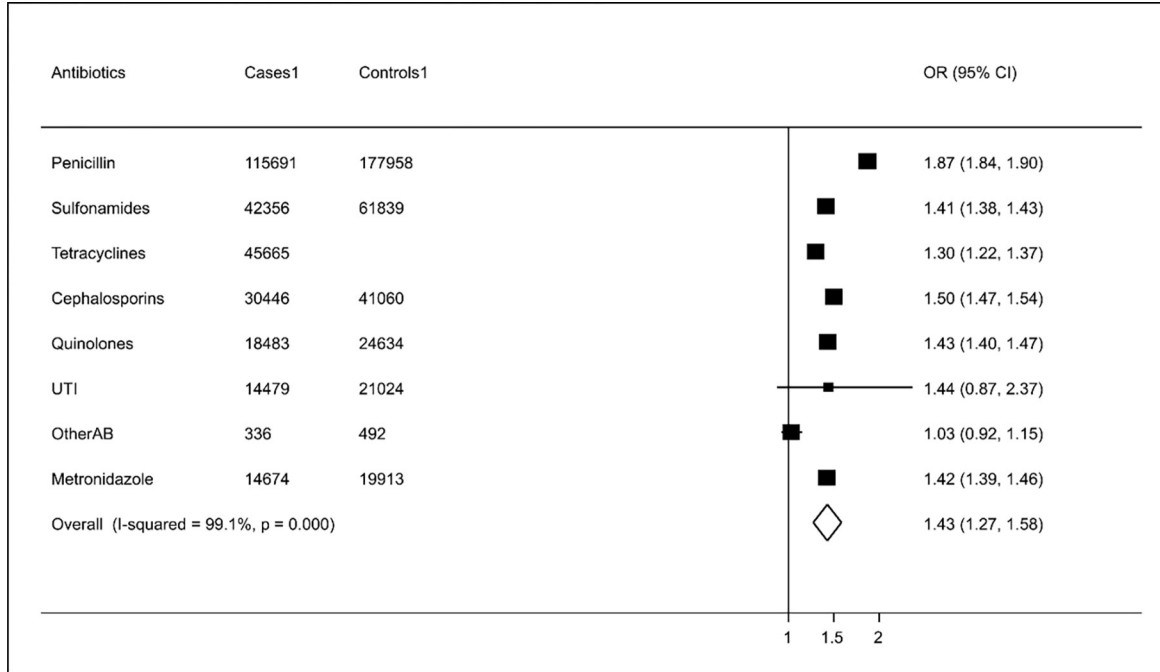

**Fig 5. Results of adjusted logistic regression showing association of HZ with different antibiotic types.** (UTI: antibiotics for urinary tract infection; Other AB: other antibiotics).

antibiotic prescriptions given in hospital nor whether patients actually consumed their pre-scribed antibiotics. A further limitation is that while the study sample is representative of the UK population, the findings may not necessarily apply to other countries with different patterns of antibiotic prescription and populations which may be more or less susceptible to anti-biotic damage.

The main limitation of the study, as with any case-control design, is possible confounding which prevents assuming causality. Some potential confounders were addressed by adjusting for several co-variates that could have influenced both prescribing and caseness. For example, HZ is more common in diabetes and rheumatoid arthritis so these were added as covariates [4, 45] as were corticosteroids given their known effect on immunosuppression. Even so, an alter-native explanation for the association of antibiotics with HZ is the role of being 'run down' or having general malaise. There is some evidence that stress makes HZ more likely [46–48] and presenting general malaise to the doctor might be more likely to attract an antibiotic prescrip-tion. Yet while this remains an alternative explanation, the timelines revealed in this study would require general malaise or stressors over ten years ago being managed with an antibiotic and for that malaise to continue to influence the appearance of HZ a decade later without sig-nificant additional antibiotics in that period. Residual confounding, however, still remains a possibility. Further, our data did not allow us to directly test the proposed mechanisms (e.g., microbiome dysbiosis), and future studies are needed to confirm a direct link between antibi-otic prescribing with microbiome dysbiosis in patients with HZ.

In summary, given the impaired immune response that allows the HZ virus to reactivate, this study has shown through an association between HZ and prior prescription of antibiotics that the latter may be involved in that reactivation. That effect, moreover, may be relatively long term. The dysbiosis that antibiotics can cause in the microbiome together with the importance of the microbiome in immune regulation is a possible pathway through which this relationship could be mediated. Any indirect effect of antibiotics on the immune response may be particu-larly important given the considerable variation in the response of individuals to viral infection.

## Supporting information

**S1 Appendix. Herpes zoster codes.**
(DOCX)

## Author Contributions

**Conceptualization:** David Armstrong, Alex Dregan, Mark Ashworth, Patrick White.

**Data curation:** Alex Dregan.

**Formal analysis:** Alex Dregan.

**Methodology:** David Armstrong, Alex Dregan.

**Project administration:** David Armstrong.

**Writing – original draft:** David Armstrong, Alex Dregan.

**Writing – review & editing:** David Armstrong, Alex Dregan, Mark Ashworth, Patrick White.

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
