## [Decision Letter · Decision Letter 0]

20 Jan 2022

PONE-D-21-26046

The effect of antibiotics on the immune response: a case control study of Herpes zoster incidence

PLOS ONE

Dear Dr. Armstrong,

Thank you for submitting your manuscript to PLOS ONE. After careful consideration, we feel that it has merit but does not fully meet PLOS ONE’s publication criteria as it currently stands. Therefore, we invite you to submit a revised version of the manuscript that addresses the points raised during the review process.

The manuscript need substantial improvement to be published.In case control studies it is essential that cases and matched controls have the same length of period in which exposure is assessed however this is not clear from the paper. The authors must specify this issue, if cases and matched control have  different  length of period in which exposure is assessed the results may be biased.

Moreover, the authors must explain why they chose as exposure the time since first antibiotic use and not the time since last antibiotic use and provide an explanation of the results obtained from a biological point of view.

We look forward to receiving your revised manuscript.

Kind regards,

Lorenza Scotti, PhD

Academic Editor

PLOS ONE

https://journals.plos.org/plosone/s/file?id=ba62/PLOSOne_formatting_sample_title_authors_affiliations.pdf”

Reviewers' comments:

Reviewer's Responses to Questions

**Comments to the Author**

1. Is the manuscript technically sound, and do the data support the conclusions?

Reviewer #1: Yes

Reviewer #2: Partly

2. Has the statistical analysis been performed appropriately and rigorously? 

Reviewer #1: No

Reviewer #2: Yes

3. Have the authors made all data underlying the findings in their manuscript fully available?

Reviewer #1: Yes

Reviewer #2: Yes

4. Is the manuscript presented in an intelligible fashion and written in standard English?

Reviewer #1: Yes

Reviewer #2: Yes

5. Review Comments to the Author

Reviewer #1: The paper studies a possible effect of antibiotics and the onset of Herpes zoster. The study design is a case-control. The paper is well written but I have several major reviews.

1. How is possible that an use of antibiotics can be associate to onset Herpes zoster after 10 year ? Maybe is better to study an acute effect of antiobiotics ?

2. I read that the matching variables are age, sex, family practice and index date for HZV. Why did authors not adjust for cohort date entry? Are they sure that cases and controls have the same lenght of follow-up ?

3. In the title there is the word ‘incidence’. How is it possible to calculate ‘incidence’ with a case control study?

4. I saw, in each figure, the “overall effect”. How di authors calculate it ? I think with a meta-analysis model (indeed in figure 5 the I2 is showed). Is it possible to include this methods in the “statistical analysis” section ?

MINOR REVIEWS

1 I would move the sentence “..Since the findings….these results here.” from “statistical analysis” to “Results”.

2 There is only a paranthesis in line 97 section “Statistical analysis”.

Reviewer #2: This is an interesting work that continues other research on the association between antibiotic use and specific pathologies.

I have some comments:

1) In the introduction, when you type "A recent meta analysis…..4 milion cases of HZV" you must insert the corresponding reference and write: "A recent meta analysis…..4 milion cases of HZV (ref Marra et al)…."

2) In Fig. 1 you write that the study involves 175335 patients with HZ and 313647 patients without EZ. However, in Tab 1 I read 163754 subjects with HZ and 331559 controls. Can you explain this discrepancy?

3) In Tab. 1 there is an empty line corresponding to the asthma variable. Are they missing values or zero values?

4) In Tab 1 I read for example LES 509 (0) where zero would represent the ratio 509/163754. I would recommend replacing the value 0 with a more correct <1 or << 1

5) In Fig. 5 you report the value of I2 without any comment in the text. If you report the value of I2 with the associated probability, you should clearly specify what these values entail

6)The last sentence “The effect of antibiotics on the immune..” appears to be little justified in the context and therefore superfluous.

6. PLOS authors have the option to publish the peer review history of their article (what does this mean?). If published, this will include your full peer review and any attached files.

Reviewer #1: No

Reviewer #2: No

---

## [Decision Letter · Decision Letter 1]

15 Sep 2022

PONE-D-21-26046R1Antibiotic driven dysregulated immune response: a case control study of Herpes zosterPLOS ONE

Dear Dr. Armstrong,

Thank you for submitting your manuscript to PLOS ONE. After careful consideration, we feel that it has merit but does not fully meet PLOS ONE’s publication criteria as it currently stands. Therefore, we invite you to submit a revised version of the manuscript that addresses the points raised during the review process. Please find below the full editorial and review comments.

We look forward to receiving your revised manuscript.

Kind regards,

Oana Săndulescu

Academic Editor

PLOS ONE

Journal Requirements:

Additional Editor Comments:

First, let me present my apologies for the delayed response that you are receiving on this manuscript. I have noticed that the manuscript was initially submitted in Aug 2021. However, I have only recently been invited as editor, this month, and I have had the chance to look through the previous peer-review process already performed. Therefore, I am now sending back the manuscript to you, with further comments from one of the previous reviewers, but I am also attaching my own comments, since I believe this is potentially a very important publication, in a less studied field, and I am listing below some of my queries that I feel could offer even better clarity to the readers:

- The portion of the title: “Antibiotic driven dysregulated immune response” and the short title “The effect of antibiotics on the immune response” should be revised as they do not reflect the actual content of the current analysis. Immune response has not been evaluated in the current study. Please revise for consistency with the research study you have performed.

- Throughout the text, when mentioning CIs, please refer to them as 95%CIs.

ABSTRACT:

- Please revise “previous varicella infection” to “previous varicella zoster virus infection”

- Please mention the measurement unit i.e., years, here: “18 to 50”

- “this study has shown that antibiotics may be involved in the reactivation of the varicella virus through their effect on the immune system”. This is not a direct conclusion of the study as you have not directly studied causality and neither immune responses. Please revise for consistency with the actual data that you presented.

INTRODUCTION:

- Is it quite uncommon to use the name “Herpes zoster virus infection” and the abbreviation HZV. More commonly, the term “varicella zoster virus” is used, abbreviated as VZV, while the disease per se is abbreviated as HZ.

- Please revise “varicella reactivation” to “varicella zoster virus reactivation”.

- “A test of this hypothesis occurs when a patient is given broad spectrum antibiotics for some unrelated condition” – please revise this statement in the text. This does not provide confirmatory proof that this hypothesis is absolutely true.

METHODS:

- “established medical diagnostic codes” is quite vague. Please clarify at least which coding system was used, if not the actual codes used for the database query.

- “the number of antimicrobial prescriptions issued during the follow-up period (i.e., 0, 1, 2, 3, 4, 5, 6-10, >10 years)” If this represents “the number”, then why is the measurement unit in years? Please clarify in the text that you are referring to the number of antibiotic prescriptions broken down by follow-up periods.

- Please revise “HIV” to “HIV infection”.

- Why was the 12 months range chosen here instead of a shorter range? “Prescriptions issued 12 months before the diagnosis were excluded from the analysis to reduce the risk of protopathic bias.”

RESULTS:

- Please check decimal points for CIs here: “aOR of 1.30 (CIs 122-137) to penicillins with an aOR of 1.87 (CIs 184-190)”.

- From Fig 1 it isn’t entirely clear when matching was performed, before or after the first exclusionary step when 27848 cases of HZ were excluded because of missing data.

- Also, in Fig 1, it is hard to follow and understand the numbers in the following box: “69420 excluded because missing data or cases with missing data/55659 no cases”. You are probably referring to controls excluded because you had to also exclude cases? Maybe a table legend could clarify this for the readers.

- Table 1: Please reconsider the title of the column “Herpes HZV”, potentially change it to “Herpes zoster”.

- Table 1: Mean and SD are generally reported with the ± sign, to avoid confusion with median (IQR).

- Fig 2: The figure title stated “previous 10+ years” while the graph lists 10 years. Please revise where appropriate, for consistency.

- The results from Fig 3 should be discussed in more depth, since the ORs are gradually decreasing with more recent antibiotic treatment. This is somewhat counterintuitive, and the potential underlying mechanisms should be discussed into more depth. Furthermore, if not accurately clarified in the manuscript’s text, it could also be inadvertently misinterpreted as: prescribing antibiotics more recently could decrease the risk for HZ.

- Fig 5: Please give more information on antibiotics for UTIs. Have you classified here antibiotics with local action only in the lower urinary tract, i.e., fosfomycin, nitrofurantoin, or also antibiotics with systemic action used for upper UTIs?

DISCUSSION:

- The explanation that you have provided to the reviewer’s prior query: “The authors must explain why they chose as exposure the time since first antibiotic use and not the time since last antibiotic use” should be added to the manuscript.

- More in depth discussion is needed for the following statement: “the finding that a first prescription more than 10 years prior to the HZV episode carried the highest risk suggests that any effect of antibiotics was long term rather than short term.” First, it would be important to justify from a pathophysiological stand point why one antibiotic course prescribed 10 years previously would still be expected to have an impact on the current microbiome. Second, when looking only at the first prescription, there is the bias of missing out on subsequent prescriptions which might be more recent and more relevant for a new onset dysmicrobism as a trigger for HZ.

- Furthermore, you state yourself that following antibiotic use “some recovery seems to occur within a year or so in adults”, which does not fit well with the observation of higher risk with older prescriptions, unless more clearly explained.

- “The results of this study add support to the view that microbiome dysbiosis (in this case caused by prior antibiotics) has an effect on the immune response”. Actually, it does not, because you have not studied the underlying immune mechanisms and you have not proven causality and the pathophysiological pathway. Please revise to a more conservative tone, keeping in line with your study’s exact findings.

- “this study has shown through an association between HZV and antibiotics that the latter may be involved in that reactivation through their effect on the immune system”. It does not. Please see my comment above and rephrase.

- Please add a paragraph discussing your study’s limitations, i.e., not being able to assess causality, underlying mechanisms, degree to which results are generalizable and to whom, etc.

Reviewers' comments:

Reviewer's Responses to Questions

**Comments to the Author**

1. If the authors have adequately addressed your comments raised in a previous round of review and you feel that this manuscript is now acceptable for publication, you may indicate that here to bypass the “Comments to the Author” section, enter your conflict of interest statement in the “Confidential to Editor” section, and submit your "Accept" recommendation.

Reviewer #1: (No Response)

2. Is the manuscript technically sound, and do the data support the conclusions?

Reviewer #1: Partly

3. Has the statistical analysis been performed appropriately and rigorously? 

Reviewer #1: No

4. Have the authors made all data underlying the findings in their manuscript fully available?

Reviewer #1: No

5. Is the manuscript presented in an intelligible fashion and written in standard English?

Reviewer #1: Yes

6. Review Comments to the Author

Reviewer #1: Althought the authors replied to the reviewers comments but I still have strong doubts.

1) In the "point by point" the authors write “…the reference time period for our study was from time of HZV diagnosis (controls were allocated the index date for cases) backward to age 18 (e.g., lifetime time period) in both cases and controls ..” this explanation does not fit with what is written in the manuscript.

2) The authors stated in the “point by point” to use the extension of case-control of Keogh et al but the "method section" of the paper does not reflect the statistical analysis of the cited paper.

3) In the "point by point" the authors stated the type of matching is “frequency” however in the paper in the line XX the auhtors wrote that the matching is “individual”. Moreover, in the new version of the manuscript they report “Incidence density sampling was used to select controls to minimise differential loss to follow-up” however if this is a case-control study, there is no follow-up period but a look-back period.

4) The model used for the meta-analysis is not clear. Did the authors use a fixed or random model ?

7. PLOS authors have the option to publish the peer review history of their article (what does this mean?). If published, this will include your full peer review and any attached files.

Reviewer #1: No

---

## [Author Response · Author response to Decision Letter 1]

11 Oct 2022

We have now responded to the editorial and reviewer comments in our document 'Response to reviewers'.

---

## [Editor Report · Decision Letter 2]

14 Oct 2022

Prior antibiotics and risk of subsequent Herpes Zoster: a population-based case control study

PONE-D-21-26046R2

Dear Dr. Armstrong,

We’re pleased to inform you that your manuscript has been judged scientifically suitable for publication and will be formally accepted for publication once it meets all outstanding technical requirements.

Kind regards,

Oana Săndulescu

Academic Editor

PLOS ONE

Additional Editor Comments (optional):

I thank the authors for addressing the previous comments from reviewers and editor.

---

## [Editor Report · Acceptance letter]

19 Oct 2022

PONE-D-21-26046R2 

Prior antibiotics and risk of subsequent Herpes Zoster: a population-based case control study 

Dear Dr. Armstrong:

I'm pleased to inform you that your manuscript has been deemed suitable for publication in PLOS ONE. Congratulations! Your manuscript is now with our production department. 

Kind regards, 

on behalf of

Dr. Oana Săndulescu 

Academic Editor

PLOS ONE